# Global economic exposure to climate change amplified by spatially compounding climate extremes

Bianca Biess ⬭ ✉, Lukas Gudmundsson ⬭ & Sonia I. Seneviratne ⬭

Despite growing evidence that climate extreme events can significantly affect local economies, the implications of cross-regional and planetary-scale dependencies in climate extremes remain inadequately understood. We demonstrate a crucial link between the projected increase in spatially compounding hot, wet, and dry extremes and the amplification of global economic exposure. Based on Earth System Model projections from the 6th phase of the Coupled Model Intercomparison Project, we analyze how planetary-scale and cross-regional dependencies can exacerbate regional disparities in economic exposure. Our findings reveal that regions with lower present-day economic wealth are more likely to face extreme events simultaneously with other areas, amplifying the potential threats to their economic stability. This study highlights the necessity of considering economic exposure to climate extremes beyond local scales, emphasizing the need for assessing cross-regional exposures and understanding the connections between localized exposures and global economic dynamics.

Over the past two decades, weather-related disasters—such as droughts, floods, heatwaves, and storms—have caused annual global gross domestic product (GDP) losses averaging 0.14%[1], with 0.05-0.82% of global GDP attributable to anthropogenic climate change[2]. Storms contributed the largest share of these losses (64%), followed by heatwaves (16%), floods (10%), and droughts (10%)[2]. In high-income countries, more than 80% of weather and climate extremes have resulted in economic losses amounting to less than 0.1% of the respective GDP, with no disaster causing losses exceeding 3.5% of GDP[3]. In contrast, in low-income countries, several disasters had economic impacts ranging from 5% to 30% of their GDP[3], highlighting the significant spatial disparities in the economic risks posed by climate change. Projections based on temperature and precipitation means, extremes, and variability suggest that global aggregate losses could reach 10% of GDP under 3 °C warming, with the most severe impacts—up to 17% of GDP—occurring in poorer, low-latitude countries[4]. Income reductions of up to 19% within the next 26 years are projected, regardless of future emission pathways[5]. Global economic damages are projected to increase non-linearly with global warming, with

significant regional variation in economic losses exacerbating global disparities further[4–7]. However, recent literature focuses on projections of economic losses due to an increase in intensity and frequency of hot or wet extreme events only. Besides an increase in individual extreme events, the simultaneous occurrence of climate extremes at different locations—referred to as "spatially compounding climate extremes"—will also likely increase due to global warming[8–10].

Although mounting evidence suggests that extreme events impact the economy at both regional and global scales, it is unclear how spatially compounding effects of these events affect economic outcomes. Understanding these dynamics is essential because they can trigger cascading disruptions across geographies and sectors, leading to globally simultaneous impacts[11,12]. When multiple regions face climate-related disruptions simultaneously, cascading effects can amplify economic instability beyond the sum of individual impacts, posing a systemic risk to the global economy due to the interconnected nature of trade, finance, and supply chains. Such systemic risks are particularly concerning in critical sectors such as food production, where disruptions can escalate global economic instability[13].

Institute for Atmospheric and Climate Science, Department of Environmental Systems Science, ETH Zurich, Zürich, Switzerland.
✉ e-mail: bianca.biess@env.ethz.ch

Thus, assessing how one region's exposure to an extreme event aligns with simultaneous exposures in other regions is crucial for understanding the resilience of the global economy to widespread climate-induced disruptions.

This study demonstrates the relevance of linking the projected increase in spatially compounding climate extremes to changes in economic exposure. We address this question by demonstrating how cross-regional and planetary-scale dependencies may further amplify regional disparities in economic exposure to extreme events. In addition to projections in hot and wet extremes, we also consider economic value affected by water scarcity and soil moisture droughts, which have not yet been analyzed on a global scale. Furthermore, we incorporate future GDP projections based on five distinct Shared Socioeconomic Pathway (SSP[14]) narratives, acknowledging that economic exposure to climate extremes will be shaped not only by climatic changes but also by socioeconomic transformations. To ensure consistency between socioeconomic and climatic assumptions, we link the SSP narratives with their corresponding Representative Concentration Pathways (RCPs[15]) following the established SSP-RCP framework[16]. Specifically, we consider five SSP-RCP combinations (SSP1-2.6, SSP2-4.5, SSP3-7.0, SSP4-6.0, and SSP5-8.5), which span a wide range of plausible futures in terms of greenhouse gas forcing and socioeconomic transformation.

## Results
### Spatially compounding economic exposure illustrated by heatwaves

Extreme events—defined as events such as heatwaves that occurred on average every 20 years in a preindustrial climate—are projected to affect more areas concurrently in the future (Fig. 1a). Consequently, the exposure of global annual GDP to such events is expected to increase, with the magnitude of this rise depending on the level of warming. Economic exposure is influenced not only by climate change but also by future socioeconomic transformation. To account for this, we incorporate GDP projections from five SSPs (SSP1–SSP5; Fig. 1b), allowing us to assess how evolving economic conditions influence additional GDP exposure relative to the 2001–2020 baseline under different warming scenarios. For each time step, model, and scenario, we first identify grid cells affected by an extreme event (Fig. 1d) and then extract the corresponding GDP distribution for the same year (Fig. 1e). Intersecting these two layers yields the GDP affected per scenario, model, and time step (Fig. 1f). The exposed GDP is further decomposed into two components: (i) a climate component, representing how exposure evolves due to changes in climate extremes while keeping GDP constant at 2020 levels; and (ii) a socioeconomic component, capturing the influence of GDP growth and distribution.

Global annual GDP affected by Global Concurrent Extremes (GCE; in this case heatwaves; Fig. 2a) is projected to increase markedly under enhanced global warming relative to present-day (2001–2020) conditions (Fig. 2b). In the mid-century period (2041–2060), scenarios with higher projected GDP, such as SSP5 (≈231 trillion USD in 2050; Fig. 1b), show the largest additional exposure under SSP5-8.5, amounting to 171 trillion USD (111–220 trillion USD, 5th–95th percentile), corresponding to roughly 45% (25–65%) of mid-century GDP relative to the present-day baseline. In contrast, SSP3, which projects the lowest GDP (≈134 trillion USD in 2050), results in 73 trillion USD (45–103 trillion USD) of additional GDP exposed under SSP3-7.0, representing around 37% (19–57%) of mid-century GDP. SSP2, currently considered the most likely scenario (≈159 trillion USD in 2050), shows 93 trillion USD (53–131 trillion USD) of additional exposure under RCP4.5, corresponding to approximately 37% (16–58%) of GDP. Across all scenarios, the increasing GDP exposure is largely driven by climate effects, with socio-economic growth contributing only marginally.

In the following, we focus on describing the additional GDP exposure under SSP2-4.5 in the mid-century period, which represents the most likely future development; results for other SSP-RCP scenarios are provided for reference in Supplementary Figs. 4–29. We introduce the concept of Regional Concurrent Extremes (RCE), which captures the fraction of regional GDP concurrently affected by extreme events occurring within the same year (Fig. 2c). In the mid-century period under SSP2-4.5, considerable regional disparities are apparent in additional GDP exposure to heatwaves (Fig. 2d). Mid-latitude regions, including the Mediterranean (MED) and Eastern Central Asia (ECA) regions, experience the highest additional exposure, reaching up to 45% (39–52%) of regional GDP relative to present-day conditions. Elevated RCE also affects tropical and subtropical regions, such as the South American Monsoon (SAM) region and southern Africa. While climate-driven changes remain the primary contributor across all regions, GDP growth increasingly aligns with heatwave-prone areas. In East Southern Africa (ESAF), approximately 10% of regional GDP exposure can be attributed to socioeconomic factors under SSP2-4.5, with even larger contributions (up to 20%) in Northern Australia (NAU) and Southeast Asia (SEA) (Supplementary Fig. 3).

Economic impacts of extreme events are inherently systemic, arising from the interconnectedness of economies through international trade, finance, and supply chains. Thus, analyzing economic exposure to extreme events requires a comprehensive perspective that accounts for both Local effects of Global Concurrent Extremes (L-GCE) and Cross-Regional Concurrent Extremes (C-RCE). Examining local GDP affected in relation to global occurrences (L-GCE; Fig. 3a) provides insights into economic resilience to spatially compounding events, while analyzing affected cross-regional GDP (C-GCE; Fig. 3c) helps disentangle regional interdependencies.

Regions with high L-GCE correspond to those projected to experience strong RCE, including MED, ECA, the Arabian Peninsula (ARP), as well as tropical regions in South America and Africa (Fig. 3b). Under SSP2-4.5, up to 108 trillion USD (66–152 trillion USD) of global GDP is concurrently affected in the mid-century period. These hotspots are frequently affected by heat extremes concurrently with other regions, reflecting their strong involvement in spatially compounding events.

Regions that are frequently affected by heat extremes concurrently with other parts of the world show high C-RCE values, highlighting the interconnected nature of global exposure (Fig. 3d). ECA, for example, exhibits selective but strong connectivity, with its highest C-RCE (around 56%) concentrated in a few key partners such as Central and Northern Australia (CAU, NAU) and tropical to southern South America. This indicates that while ECA is highly exposed, its concurrent exposure mainly occurs through specific teleconnections with a few key partner regions rather than through connections with many regions worldwide. In contrast, MED functions as a global connectivity hub, maintaining high concurrency with a broad set of regions across multiple continents—including North America, Australia/New Zealand, Southern South America, and large parts of Africa. This extensive linkage amplifies MED's already elevated regional GDP exposure to heatwaves.

In summary, regional disparities in GDP affected by heatwaves are already pronounced in the mid-century and can be further amplified by elevated simultaneous exposure to heatwaves across multiple regions. For regions with high regional exposure, cross-regional exposure acts as a multiplier, linking local economic shocks to global systemic risks. Some regions show concentrated connectivity with a few key partners, whereas others, like MED, are highly concurrent with many regions across continents, highlighting the spatial complexity of economic exposure to heat extremes.

### GDP affected by spatially compounding heavy precipitation

In the mid-century period, global GDP exposed to heavy precipitation events (GCE; Fig. 4a) is projected to reach 32 trillion USD under SSP5-

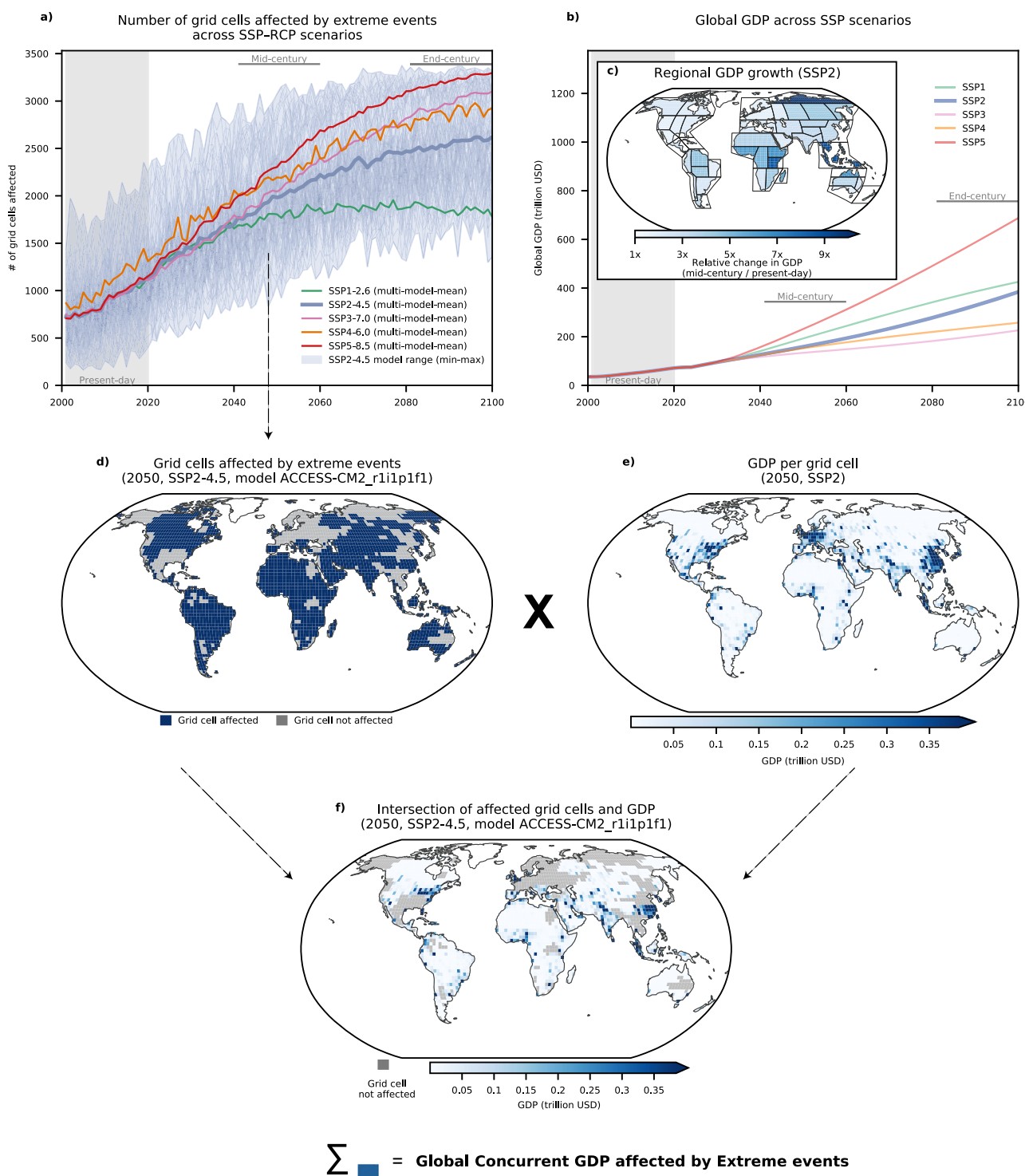

**Fig. 1 | Assessing economic exposure to climate extremes.** Illustration of the workflow for quantifying the share of global gross domestic product (GDP) exposed to climate extremes, exemplified for heatwaves in the year 2050 under Shared Socioeconomic Pathway 2 with a radiative forcing level of 4.5 W m⁻² (SSP2-4.5). **a** Climate models project an increasing number of grid cells affected by heatwaves. **b** Global GDP trajectories are shown for five Shared Socioeconomic Pathways (SSPs[14]), using projections from Wang and Sun[48]. **c** Regional changes in GDP under SSP2 for the mid-century period (2041–2060) relative to the present-day conditions (2001–2020), highlighting spatially heterogeneous economic growth patterns. **d** Grid cells affected by a heatwave in 2050, illustrated for one example climate model (ACCESS-CM2, ensemble member r1i1p1f1). **e** Spatial distribution of GDP per grid cell in 2050 under SSP2. **f** The intersection of the grid cells affected by a heatwave (**d**) with the GDP distribution (**e**) identifies the GDP exposed to heatwaves in 2050; summing these globally yields the concurrent global GDP exposed, exemplified here for SSP2-4.5 in 2050 for one model.

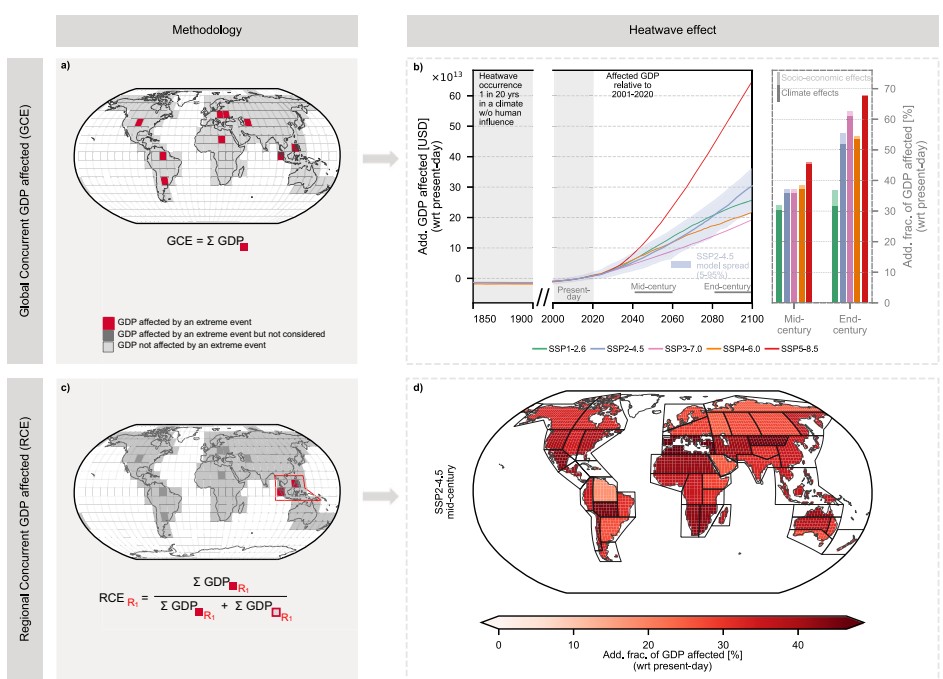

**Fig. 2 | Global and regional exposure of gross domestic product (GDP) to heatwaves. a, b** show the Global Concurrent GDP affected (GCE). **a** Methodology: The GDP of all grid cells experiencing an extreme event is summed globally, and the GCE is expressed relative to present-day climate conditions. **b** Results: Trajectories of additional GCE under five Shared Socioeconomic Pathway (SSP) narratives combined with their respective Radiative Concentration Pathway (RCP). The bar plot shows the additional fraction of GDP affected by heatwaves in the mid-century (2041–2060) and end-century (2081–2100) periods for each scenario, distinguishing between GDP exposure driven by climate change (present-day GDP becoming newly affected) and exposure driven by economic growth (additional GDP beyond 2020 levels becoming affected). **c, d** show the Regional Concurrent GDP affected (RCE). **c** Methodology: For each region defined in the Intergovernmental Panel on Climate Change (IPCC) Sixth Assessment Report (AR6 regions[46]; see Supplementary Fig. 1), the GDP of affected grid cells is summed and normalized by the time-varying absolute GDP level. **d** Results: Regional distribution of the additional fraction of GDP affected by heatwaves in the mid-century period under Shared Socioeconomic Pathway 2 combined with a radiative forcing level of 4.5 W m$^{-2}$ (SSP2-4.5).

8.5 (13–57 trillion USD; 5th–95th percentile), equivalent to about 6.4% of global GDP. Under SSP3-7.0, exposed GDP amounts to 13 trillion USD (4–24 trillion USD; ≈4.5% of GDP), while SSP2-4.5 yields an intermediate estimate of 16 trillion USD (6–30 trillion USD; ≈4.5% of GDP). In all scenarios, climate effects drive the majority of exposure, although socio-economic factors contribute up to ≈0.5–1% of total exposure, reflecting GDP growth concentrated in regions particularly prone to heavy precipitation.

Significant regional disparities emerge (RCE; Fig. 4b), with tropical regions projected to experience the largest increases: in the SSP2-4.5 mid-century scenario, relative exposure reaches about three times the global average. In several tropical regions— particularly Northern South America (NSA), Central Africa (CAF), South Asia (SAS), and SAM—economic growth in precipitation-prone areas further amplifies exposure. Socio-economic effects dominate in NSA and contribute up to one-fifth in other regions, while climatic effects remain the primary driver overall (Supplementary Fig. 3).

Notably, these regions also exhibit high concurrent exposure to global events (L-GCE; Fig. 4c), potentially amplifying economic impacts through spatially compounding effects. The highest L-GCE arises when the spatial compounding effect of tropical African regions is considered together with concurrent impacts across the rest of the world. Under the SSP2-4.5 mid-century scenario, a heavy precipitation event in these regions would result in approximately 18 trillion USD being concurrently affected on a global scale. A substantial share of these elevated local effects is driven by intra-continental C-RCE (Fig. 4d) and its overlaps with tropical South American regions— Northeastern South America (NES), Northwestern South America (NWS), SAM, and NSA— and, to a lesser extent, with selected high-latitude, boreal, and Australasian regions.

In summary, regional disparities in GDP affected by heavy precipitation are projected to widen markedly, with the strongest increases in relative exposure occurring in the tropics, where extreme rainfall events become more frequent. These physical changes coincide with rapid socio-economic growth, amplifying regional exposure such that relative GDP affected reaches up to three times the global mean. At the same time, intra-continental simultaneous exposure— linking multiple African regions experiencing heavy precipitation concurrently—further heightens economic risk, underscoring the growing importance of intra-continental compounding effects.

**GDP affected by spatially compounding soil moisture droughts**
Under SSP5-8.5, global concurrent GDP affected by soil moisture droughts (GCE; Fig. 5a) is projected to reach 37 trillion USD in the mid-century period (10–75 trillion USD; 5th–95th percentile), corresponding to roughly 6.5% of global GDP (-2 to 18%), with all near-term exposure driven by climatic effects. As for heavy precipitation, SSP5 exhibits the highest absolute and relative exposure due to the strong physical intensification of extreme events. Under SSP3-7.0, the exposure is lower, amounting to 16 trillion USD (3–30 trillion USD) in the mid-century period, with relative exposure of ≈5.2% (−2 to 15%). SSP2-4.5 falls between these extremes, with absolute exposure of 20 trillion USD (5–43 trillion USD) and relative exposure of ≈5% (−2 to 15%), to which socio-economic effects contribute approximately 0.3%.

Under the SSP2-4.5 mid-century scenario, regional disparities in GDP affected by soil moisture drought intensify (RCE; Fig. 5b), with drought-prone regions reaching up to three times the global average in relative exposure and largely coinciding with areas projected to experience pronounced drying[8], including NSA, NES, SAM, and South-Western South America (SWS) in the Americas; ESAF, WSAF, and Madagascar (MDG) in Africa; and Western and Central Europe

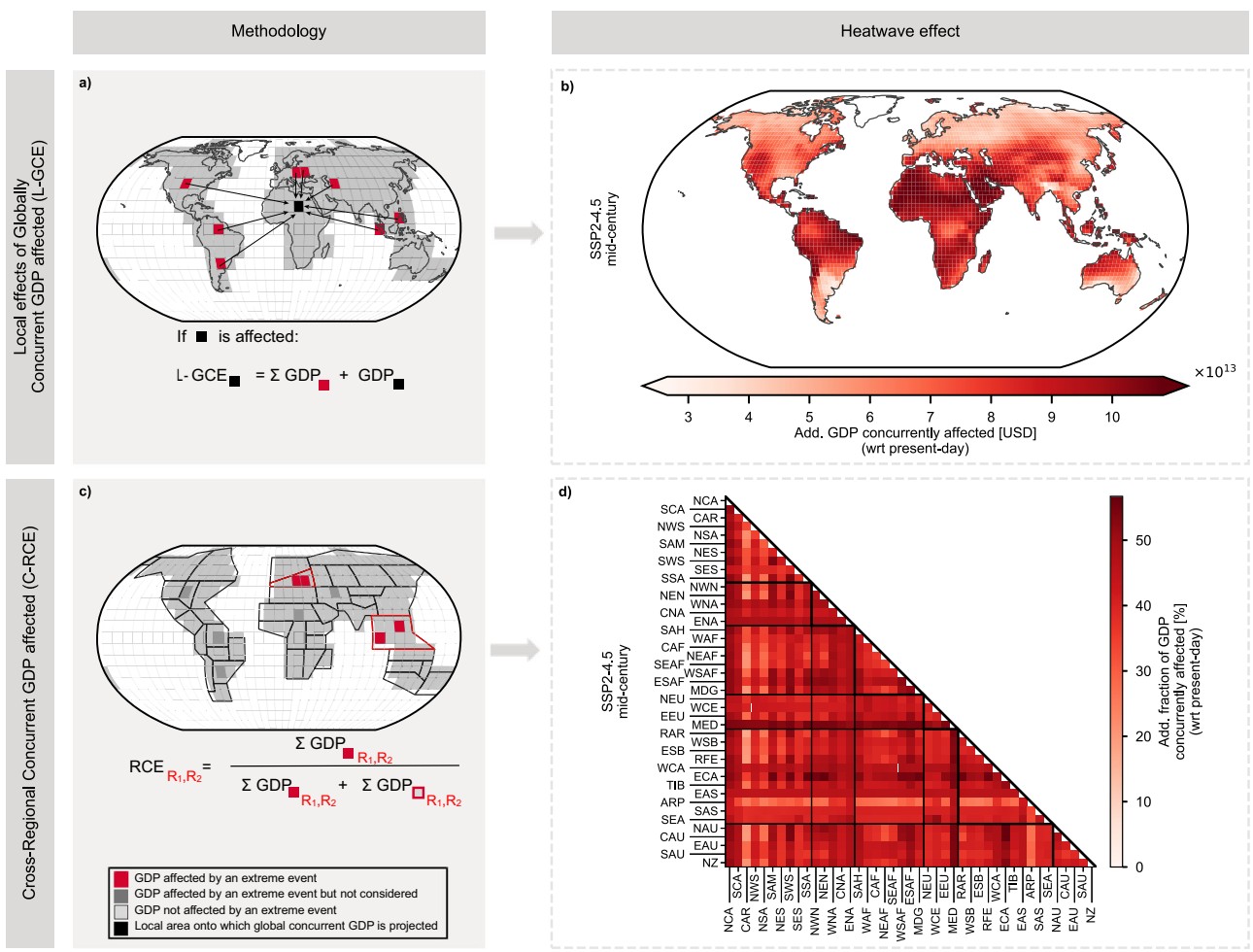

**Fig. 3 | Exposure of gross domestic product (GDP) to spatially compounding heatwaves. a, b** Show the Local Effects of Global Concurrent GDP affected (L-GCE). **a** Methodology: For each grid cell and time step, the global concurrent GDP affected (GCE) is computed and attributed to the locations simultaneously experiencing extreme events. **b** Results: Spatial distribution of additional L-GCE for heatwaves under Shared Socioeconomic Pathway 2 combined with a radiative forcing level of 4.5 W m⁻² (SSP2-4.5) in the mid-century period (2041–2060),

identifying regions where local impacts align with global economic effects. **c, d** show the Cross-Regional Concurrent GDP affected (C-RCE). **c** Methodology: Pairs of regions are analyzed to quantify the GDP affected by extreme events occurring simultaneously across both regions. **d** Results: Additional C-RCE for heatwaves under SSP2-4.5 in the mid-century period, highlighting interregional linkages of concurrent heatwave exposure.

(WCE) and MED in Europe and the Mediterranean. Although southern North America is also projected to experience drying, it does not emerge as a regional hotspot for high RCE in our analysis. Whereas for heavy precipitation events, high RCE strongly coincides with regions experiencing rapid GDP growth, enhancing relative GDP exposure, but this socio-economic amplification applies only to some drought hotspots. For example, NES and SAM both exhibit similarly high relative GDP exposure (RCE) to soil moisture droughts (≈13%), but the contributions of climate and socio-economic effects differ. NES shows moderate GDP growth (Fig. 1c), yet because it occurs in drought-prone areas, relative exposure is amplified (Supplementary Fig. 3). In contrast, SAM is projected to undergo higher GDP growth, but much of it is outside drought-prone regions, dampening socio-economic amplification.

The amplification of potential economic impacts through global compounding effects (L-GCE; Fig. 5c) remains pronounced in these regions. Under the SSP2-4.5 mid-century scenario, up to 27 trillion USD are projected to be affected concurrently on a global scale. Regions highly impacted by soil moisture droughts also exhibit substantial simultaneous exposure with other drying regions (C-RCE; Fig. 5d). For example, South American and African drying regions display exceptionally high intercontinental concurrent exposure, whereas Southern

Australia (SAU) and New Zealand (NZ) show strong concurrent exposure with the South American and European drying regions.

In summary, GDP affected by soil moisture droughts exhibits pronounced regional disparities, with drought-prone areas in South America, Africa, and the Mediterranean experiencing up to three times the global average relative exposure. While the physical intensification of droughts drives most of the GDP exposure, concentrated GDP growth in select regions further amplifies local economic exposure. These regions also show substantial simultaneous exposure with other drying areas, both within and across continents, highlighting the compounding nature of drought impacts and the potential for amplified global economic consequences.

## GDP affected by spatially compounding water scarcity

In the mid-century period under SSP5-8.5, annual global GDP exposed to water scarcity (GCE; Fig. 6a) is projected to reach 30 trillion USD (9-61 trillion USD; 5th–95th percentile), corresponding to roughly 4.8% of global GDP (−2 to 15%). Under SSP3-7.0, absolute exposure is lower, amounting to 11 trillion USD (2.5-25 trillion USD; ≈3% of GDP, −3 to 11%). SSP2-4.5 shows intermediate values, with absolute exposure of 17 trillion USD (4.4-34 trillion USD; ≈3.7% of GDP, −3 to 12%) and a comparatively smaller contribution from socio-economic growth.

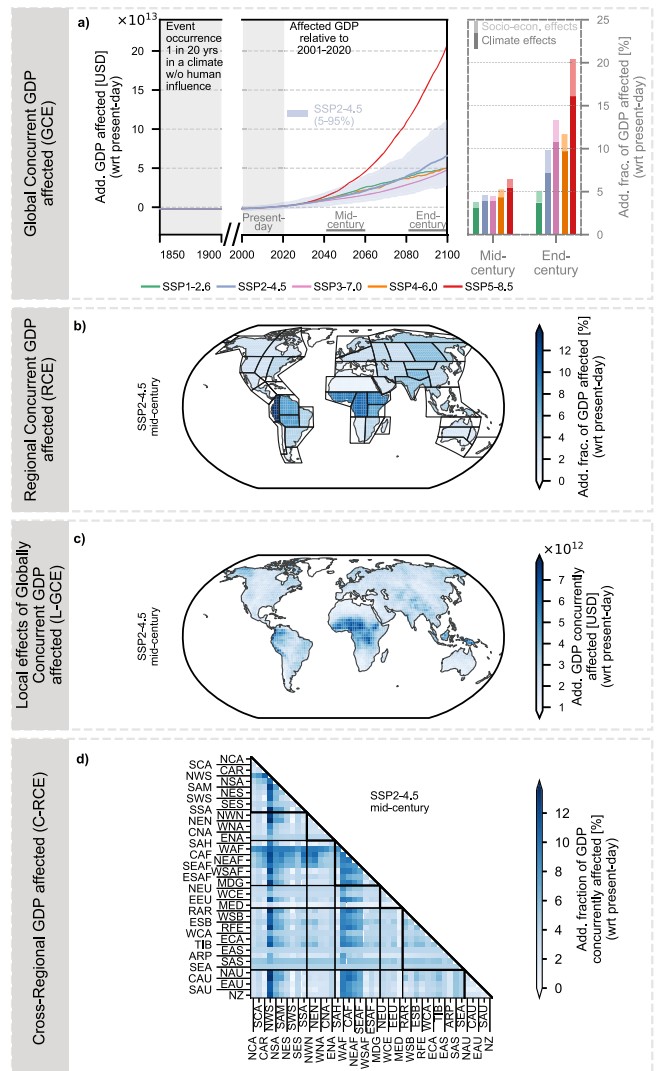

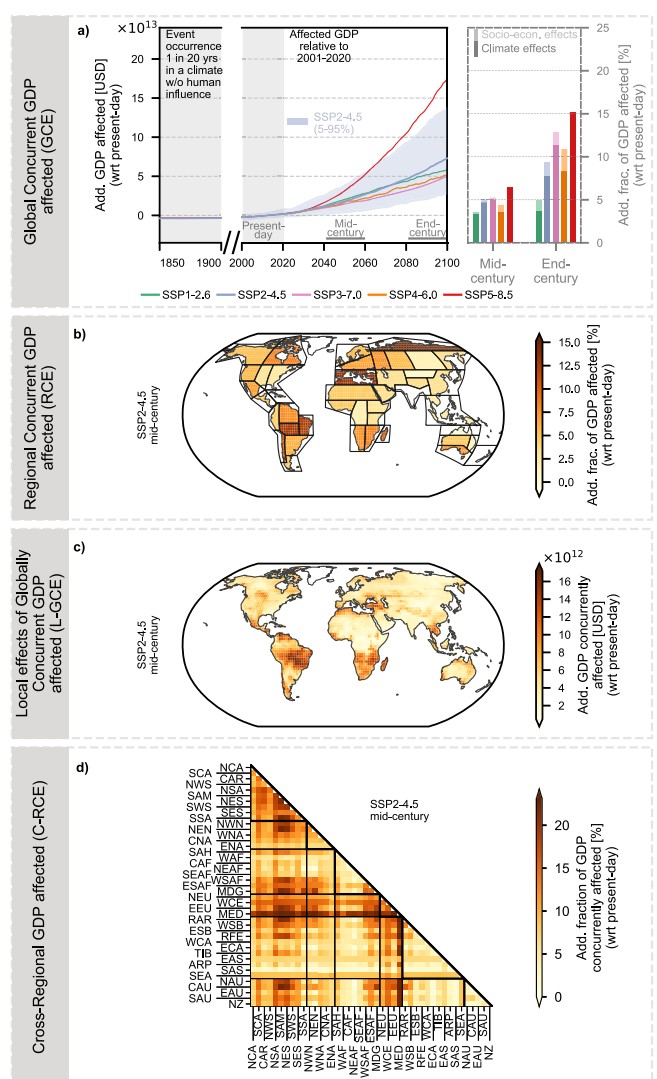

**Fig. 4 | Exposure of gross domestic product (GDP) to spatially compounding heavy precipitation, relative to present-day climate conditions (2001–2020).** **a** Global Concurrent GDP affected (GCE): Trajectories of additional GCE under five combined socioeconomic and radiative forcing scenarios (SSP-RCP). The bar plot on the right shows the additional fraction of GDP affected by heavy precipitation in the mid-century (2041–2060) and end-century (2081–2100) periods, separating exposure driven by climate change (present-day GDP becoming newly affected) from exposure driven by economic growth (GDP added after 2020 becoming affected). **b** Regional Concurrent GDP affected (RCE): Regional distribution of additional GDP affected by heavy precipitation under Shared Socioeconomic Pathway 2 combined with a radiative forcing level of 4.5 W m⁻² (SSP2-4.5) in the mid-century period. **c** Local Effects of Global Concurrent GDP affected (L-GCE): Spatial distribution of additional L-GCE under the SSP2-4.5 scenario in the mid-century period. **d** Cross-Regional Concurrent GDP affected (C-RCE): Additional C-RCE under the SSP2-4.5 scenario in the mid-century period.

**Fig. 5 | Exposure of gross domestic product (GDP) to spatially compounding soil moisture droughts, relative to present-day climate conditions (2001–2020).** **a** Global Concurrent GDP affected (GCE): Trajectories of additional GCE under five combined socioeconomic and radiative forcing scenarios (SSP-RCP). The bar plot on the right shows the additional fraction of GDP affected by soil moisture droughts in the mid-century (2041–2060) and end-century (2081–2100) periods, separating exposure driven by climate change (present-day GDP becoming newly affected) from exposure driven by economic growth (GDP added after 2020 becoming affected). **b** Regional Concurrent GDP affected (RCE): Regional distribution of additional GDP affected by soil moisture droughts under Shared Socioeconomic Pathway 2 combined with a radiative forcing level of 4.5 W m⁻² (SSP2-4.5) in the mid-century period. **c** Local Effects of Global Concurrent GDP affected (L-GCE): Spatial distribution of additional L-GCE under the SSP2-4.5 scenario in the mid-century period. **d** Cross-Regional Concurrent GDP affected (C-RCE): Additional C-RCE under the SSP2-4.5 scenario in the mid-century period.

In all scenarios, climate effects drive the majority of exposure, although socio-economic factors contribute between 1.5% (SSP2-4.5) and 8% (SP4-6.0), reflecting GDP growth concentrated in regions particularly prone to water scarcity.

Under the SSP2-4.5 mid-century scenario, regions with high relative GDP exposure to water scarcity (RCE; Fig. 6b) only partly overlap with areas experiencing pronounced drying[8,17]. Notable hotspots include NSA, SAM, SAU, and MED, with RCE around 6–7% of regional GDP (3–10%), roughly twice the global mean. Other regions, such as the Sahara (SAH), RAR, and ECA, also show

elevated exposure. Across most regions, climate effects dominate, while socio-economic growth amplifies exposure in some areas, particularly parts of SAH and RAR. In contrast, in NSA, GDP growth occurs mostly outside the most water-stressed areas, slightly moderating the overall exposure signal (Supplementary Fig. 3).

Regions such as Western and Central Africa (WAF, CAF) and SAH exhibit the most pronounced global spatial compounding effects (L-GCE; Fig. 6c), with up to 10.7 trillion USD (0-34 trillion USD) in economic value concurrently affected globally in the mid-century period. SAH, WAF, and CAF show strong links with regions across Europe,

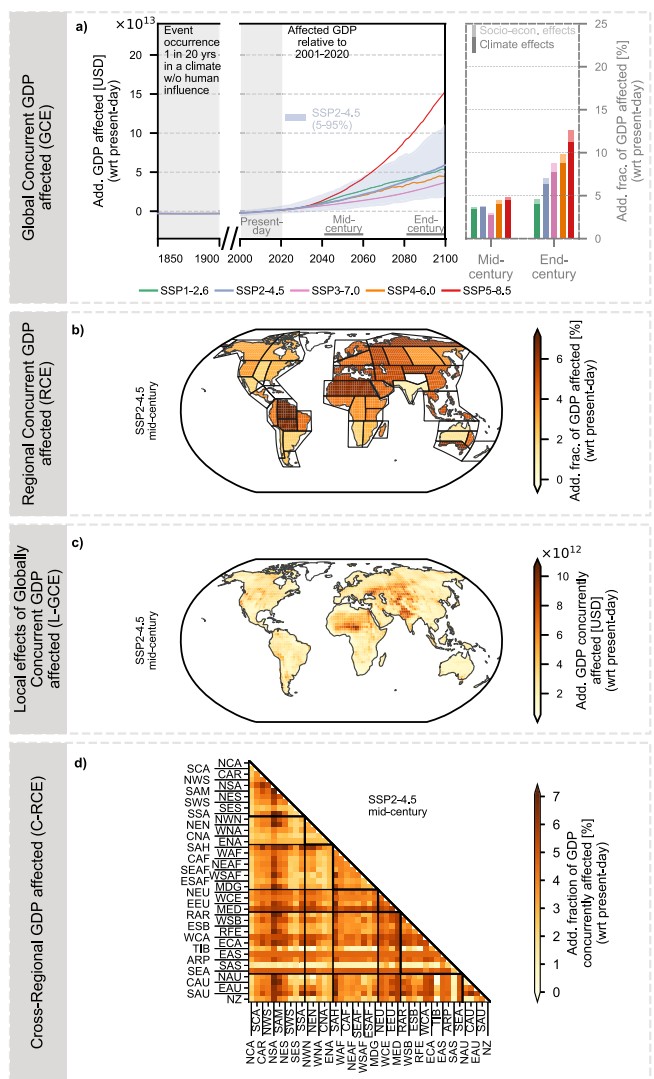

**Fig. 6 | Exposure of gross domestic product (GDP) to spatially compounding water scarcity, relative to present-day climate conditions (2001–2020).**
**a** Global Concurrent GDP affected (GCE): Trajectories of additional GCE under five combined socioeconomic and radiative forcing scenarios (SSP-RCP). The bar plot on the right shows the additional fraction of GDP affected by soil moisture droughts in the mid-century (2041–2060) and end-century (2081–2100) periods, separating exposure driven by climate change (present-day GDP becoming newly affected) from exposure driven by economic growth (GDP added after 2020 becoming affected). **b** Regional Concurrent GDP affected (RCE): Regional distribution of additional GDP affected by water scarcity under Shared Socioeconomic Pathway 2 combined with a radiative forcing level of 4.5 W m⁻² (SSP2-4.5) in the mid-century period. **c** Local Effects of Global Concurrent GDP affected (L-GCE): Spatial distribution of additional L-GCE under the SSP2-4.5 scenario in the mid-century period. **d** Cross-Regional Concurrent GDP affected (C-RCE): Additional C-RCE under the SSP2-4.5 scenario in the mid-century period.

Australasia, and tropical South America (C-RCE; Fig. 6d), highlighting their key role in cross-regional concurrent exposure, while other regions experience elevated but comparatively lower impacts.

In summary, GDP affected by water scarcity exhibits pronounced regional disparities, with some regions experiencing up to twice the global average relative exposure. Climate effects dominate GDP exposure in all regions, while socio-economic growth further amplifies exposure in select areas—particularly in parts of the Sahara and the Russian Arctic—where economic development occurs in already water-stressed regions. In contrast, in NSA, GDP growth is projected to

concentrate outside the most water-stressed areas, slightly dampening local exposure. WAF, CAF, and SAH emerge as the primary hotspots for simultaneous global exposure, exhibiting pronounced cross-regional effects across Europe, Australasia, and tropical South America. These patterns highlight the potential for compounding economic impacts of water scarcity and the combined importance of climatic and socio-economic drivers in shaping global and regional disparities.

## Discussion

Building on the existing evidence that climate extremes already impose substantial economic burdens and that these are projected to increase with global warming, our study shifts the focus from realized economic damages to GDP exposure to climate extremes, with particular emphasis on spatially compounding events, where multiple regions experience extremes simultaneously. By estimating GDP exposure, we capture a higher magnitude of potentially affected economic value and can identify not only where individual extremes may occur, but also where their co-occurrence across regions could amplify economic risks. We analyze each of the four extreme event types independently to highlight the different ways individual hazards may affect future economic conditions. At the same time, some extremes are physically correlated—for example, soil-moisture-temperature feedbacks[18] may link hot and dry conditions—thereby, exposure associated with different hazards should not be interpreted as additive. To account for potential overlaps, we additionally quantify GDP exposed to at least one of the four extreme event types (Supplementary Fig. 4).

Our approach addresses a critical gap in the literature regarding dry extremes, whose economic impacts are often indirect, long-term, and under-quantified[19,20]. Despite data limitations[20,21], our analysis provides insights into how droughts interact across regions to generate spatially compounding exposure, highlighting hotspots where global GDP is most likely to coincide with multiple simultaneous extremes. This perspective underscores the combined importance of climatic and socio-economic drivers in shaping future global and regional economic vulnerabilities.

The pronounced regional disparities in projected GDP exposure across the four extreme event types examined here can be understood through the physical mechanisms that govern the formation and intensification of these extremes. Pronounced increases in GDP exposed to heatwaves in mid-latitude and tropical regions are coherent with the physical mechanisms that intensify heat extremes. Strong land-atmosphere feedbacks in mid-latitude transitional regimes[8] and the heightened sensitivity of tropical regions with low interannual temperature variability[22–24] render these areas particularly susceptible to intensifying heat extremes. As mean temperatures rise, even small additional warming in the tropics leads to more frequent exceedances of critical thresholds[21], resulting in substantial increases in the share of economic activity exposed to extreme heat. While exposure does not directly translate into realized economic losses, it increases the likelihood of adverse outcomes. Socio-economic risks are particularly pronounced in low-income tropical regions, where vulnerability is high and coping capacity remains limited[4,5,21]. Moreover, many regions are projected to face compounded challenges due to simultaneous exposure across multiple areas, driven by global warming and the rising co-occurrence and spatial extent of dependent temperature extremes[10]. Recent heatwaves illustrate how short-lived but intense thermal extremes can accumulate into long-term macroeconomic effects. Repeated exposure to extreme heat has been shown to depress growth, with the lowest income decile losing around 8% of GDP per capita annually—more than double the 3.5% observed in the wealthiest decile[21]. Losses stem primarily from reduced labor productivity, crop damage, and higher mortality, especially in agriculture, construction, and energy sectors[1].

A similar pattern emerges for GDP exposed to heavy precipitation events, particularly in tropical regions. Thermodynamic and dynamic

mechanisms, such as enhanced atmospheric moisture and intensified convection under warming, drive increases in heavy precipitation. These processes amplify heavy precipitation more strongly in the tropics than in extratropical regions[8,10,25,26]. Beyond these physical drivers, socioeconomic factors further reinforce disparities in exposure, as projected economic growth increasingly coincides with regions prone to heavy precipitation extremes. Floods, often triggered by heavy precipitation events[27], have historically caused substantial economic damage across many continents, with Oceania, Asia, and Europe experiencing the largest absolute losses[1]. Subnational evidence shows that extreme rainfall and wet days can depress growth, particularly in services and manufacturing sectors in high-income countries[28]. Looking ahead, our analysis indicates that the frequency of wet extremes is projected to increase over tropical Africa. In regions where rapid economic growth overlaps with these rising precipitation risks, the potential for higher GDP exposure emerges.

Observed drying patterns in both water balance (P-ET) and soil moisture show clear regional disparities, driven by precipitation deficits, rising atmospheric evaporative demand (AED), and land-atmosphere feedbacks. Extratropical regions, such as Western and Central Europe, have experienced drying primarily from increased AED and evapotranspiration, whereas several tropical areas—including the northern Andes and central Africa—have dried mainly due to reduced rainfall[29]. Recent droughts have caused persistent GDP declines averaging -2% within four years, particularly in economies dependent on rain-fed agriculture and hydropower[1]. Looking ahead, AED is projected to increase across large tropical and subtropical regions—including southern Africa, the Amazon, the Mediterranean, and southern North America—intensifying both P-ET deficits and soil drying[30]. Soil moisture droughts, however, are expected to become even more widespread than P-ET deficits for a given level of warming. This is due to additional land-atmosphere feedbacks and surface processes[8,30]. This analysis reflects trends for regions projected to undergo pronounced soil moisture drying as of 2 °C of warming—including Western and Central North America, the Mediterranean, southern Africa, and Australia[8]—showing that the share of global and regional GDP affected by soil moisture droughts is far greater than that affected by water scarcity. Both variables, however, exhibit sharply rising concurrent exposure, highlighting the importance of spatially compounding climate extremes for future drought risk.

Besides the extreme events investigated here, many other critical climate factors may lead to potential economic losses. These include potential losses or benefits from changes in global mean temperature[7], which is expected to remain a primary driver of projected losses[4]. Other critical factors include losses from other extreme events such as tropical cyclones[31,32] and wildfires[33]. Further channels such as impacts from sea-level rise[34], tipping points in the climate system[35], as well as non-economic damages (reductions in ecosystem services or human health, effects on morbidity and mortality, loss of subjective well-being from less tangible benefits)[36,37] are not considered here. Some recent studies report on lagged temperature- and precipitation-related impacts on economic growth and GDP levels[5,7,28,38], as well as on indirect cascading and spillover effects, for example, due to supply-chain disruptions[39]. While this study does not investigate all of the possible consequences of spatially compounding climate extremes, it demonstrates their key relevance for upcoming impacts of human-induced climate change on the economy.

This study highlights the importance of considering exposure to climate extremes beyond local scales, requiring an assessment of cross-regional exposures and a deeper understanding of the links between localized impacts and global economic dynamics. Regional disparities in GDP affected by extreme events are often exacerbated by spatially compounding effects. The strengthening of spatially compounding climate extremes has significant consequences for exposed economic wealth, impacting various sectors such as humanitarian aid and insurance. For instance, the fundamental risk management strategy in the (re-)insurance industry relies on spatial risk diversification across geographies[40]. The same applies to ex-ante financial instruments like sovereign catastrophe risk pools, which have been advocated by the Sendai Framework[41] and the Paris Agreement[42] to foster more resilient ex-ante financial instruments in low- to middle-income countries. We demonstrate that global warming is projected to alter the randomness and correlation of events across regions and countries, increasing economic exposure and limiting the effectiveness of traditional risk diversification strategies. Recent research highlighted the potential for improving a country's financial resilience through global catastrophe pooling[43]. We show that, on one hand, concurrent exposure will be exceptionally high on intercontinental scales (especially regarding heavy precipitation events in low- to-middle income regions), further constraining the effectiveness of intercontinental risk pools. On the other hand, however, concurrent economic exposure to climate extremes is also projected to increase on supra-continental scales. Therefore, careful consideration and in-depth investigation should be given to which supraregional and supracontinental regions are pooled. Policy coordination and international cooperation can play an important role in addressing the challenges posed by simultaneous climate events, which may require joint recovery efforts, resource sharing, and comprehensive contingency planning. Similarly, investors and insurers may need to account for the likelihood of concurrent events across multiple regions when assessing risks and planning for financial resilience.

Overall, observed evidence already underscores stark inequalities in economic losses from climate extremes. Low-income economies have historically borne the heaviest burdens from heat and drought, while high-income regions have experienced the largest absolute damages from wet extremes, mainly affecting services and manufacturing. Although historical losses from wet extremes have been concentrated in high-income regions, projections suggest that increasing heavy precipitation in tropical regions could alter these patterns, potentially shifting the distribution of future economic impacts. Projections indicate that economic exposure to climate extremes will intensify and become increasingly interconnected. Spatially compounding climate extremes pose systemic risks that transcend regional boundaries, amplifying global vulnerabilities through trade, production, and supply chains. This highlights the importance of investigating region- and variable-specific risks, accounting for trade partners, and designing insurance schemes aligned with the physical realities of compounding extremes. Crucially, limiting global warming to +1.5 °C substantially reduces the economic value at risk, whereas warming beyond +2 °C drives dramatically higher exposure and costs. These findings reinforce the urgency of stringent climate action and coordinated risk management strategies to mitigate the escalating socioeconomic impacts of spatially compounding climate extremes.

## Methods
### Climate extreme indicators
To assess future changes in the global GDP affected by spatially compounding heatwaves, heavy precipitation, soil moisture droughts, and water scarcity, Earth System Models (ESMs) contributing to the Scenario Model Intercomparison Project[44,45] of the Coupled Model Intercomparison Project phase 6 (CMIP6) are used. We consider the first five available ensemble members of historical (1850–2014)[44] and future simulations (2015–2100) under five socioeconomic pathways paired with corresponding emission scenarios (SSP1-2.6, SSP2-4.5, SSP3-7.0, SSP4-6.0, SSP5-8.5)[14–16], giving equal weight to each model. Detailed information on the ESMs, including model name and ensemble members, is summarized for reference in Supplementary Tables 1–4. The CMIP6 data undergo centralized preprocessing, which involves interpolating model outputs to a common 2.5° grid using

second-order conservative remapping, resolving dimensional and structural inconsistencies, and excluding files with highly unrealistic values. We analyze daily maximum temperature, total daily precipitation, monthly soil moisture within the root zone (top 10 cm), and monthly precipitation−evapotranspiration (P-E) from all models and simulation runs to determine the extreme event indicators.

The selected extreme event indicators−heatwaves, heavy precipitation, soil moisture drought, and water scarcity−follow established definitions assessed in the IPCC Sixth Assessment Report[8] and were chosen because they represent well-understood manifestations of anthropogenic climate change, capturing events of high societal relevance across key sectors including health, agriculture, and infrastructure.

The indicators are characterized using a percentile-based event definition. The percentile climatology of each event type is calculated based on the 95th percentile calculated from the reference period 1850−1900. The percentile climatology, as well as the occurrence of the event type, is calculated for each grid cell of the simulated data.

The heatwave indicator is defined based on a 14-day moving average of daily maximum temperature and its subsequently extracted annual maximum. Each extracted annual 14-day temperature maximum that exceeds the 95th percentile climatology is defined as a year in which a heatwave occurs.

Heavy precipitation events are defined based on a five-day moving precipitation sum and its subsequently extracted annual maximum. Each extracted annual five-day precipitation sum that exceeds the amount of precipitation in the 95th percentile climatology is defined as a year in which a heavy precipitation event occurs.

Soil moisture drought events are defined based on monthly root-zone soil moisture sums and the corresponding annual minima. Any annual minimum that falls below the 5th percentile of the baseline climatology is considered a year in which a soil moisture drought occurs.

Years with the occurrence of water scarcity are defined based on the monthly precipitation-evapotranspiration (P-E) sums and the corresponding annual minima. Any annual that falls below the 5th percentile of the baseline climatology is defined as a water-scarce year.

### Quantification of GDP affected by climate extremes

On a grid cell level, we intersect the extracted climate extreme events with global gridded GDP datasets consistent with the Shared Socioeconomic Pathways (SSPs)[33]. The GDP datasets (in 2005 PPP USD) provide annual values for the historical period 2000−2020 and projections from 2025 to 2100 at 5-year intervals under the five SSPs. For intermediate years, values are assigned to the nearest 5-year timestep (e.g., 2026−2029 are assigned to 2025). The exposure dataset is aggregated to a 2.5° × 2.5° grid using an area-weighted sum of the high-resolution cells to facilitate comparison with ESM simulations while preserving absolute GDP values.

GDP affected is quantified at multiple scales using the following metrics: Global Concurrent GDP affected (GCE), Regional Concurrent GDP affected (RCE, by AR6 regions[46], Supplementary Fig. 1), Local effects of Globally Concurrent GDP affected (L-GCE), and Cross-Regional GDP affected (C-RCE).

The fraction of GDP affected is calculated as shown in Eq. (1).

$$\text{fractional GDP affected} = \frac{\text{GDP affected}}{\text{GDP total}} \quad (1)$$

Changes in GDP affected are expressed relative to the mean of the present-day period (2001−2020) and computed according to Eq. (2):

$$\Delta X = X(t_1) - \overline{X_{\text{present-day}}} \quad (2)$$

where $\overline{X_{\text{present-day}}}$ denotes the average over the baseline years. All subsequent changes are reported relative to this baseline, highlighting additional future effects compared to present-day levels.

To attribute changes in GDP affected, we quantify the fraction of GDP exposed to climate extremes and decompose its future changes into two main contributions: (i) climate-driven, reflecting how exposure evolves due to changing climate extremes while holding GDP constant at 2020 levels, and (ii) socio-economic-driven, capturing changes due to GDP growth, including regions where economic development coincides with stronger climate exposure[47].

For this, we define the following components, illustrated here for Global Concurrent GDP affected (GCE). GCE($t$) represents the full model, where both climate and GDP evolve simultaneously. GCE$_{\text{fixedGDP}}$($t$) captures the climate effect, representing changes in GCE due to evolving climate extremes while holding GDP constant at 2020 levels. GCE$_{\text{fixedClimate}}$($t$) captures the socio-economic effect, representing changes in GCE due to GDP growth while keeping climate extremes at 1850−1900 levels. Finally, GCE$_{\text{interaction}}$($t$) is the interaction term, which captures additional exposure in regions where economic growth coincides with more frequent climate extremes; this term cannot be directly attributed to climate or socio-economic effects alone but emerges from their joint evolution.

The total change in GCE can be decomposed as described in Eq. (3):

$$\mathbf{GCE} = \underbrace{\mathbf{GCE}_{\text{fixedGDP}}}_{\text{i) climate effect}} + \underbrace{\mathbf{GCE}_{\text{fixedClimate}}}_{\text{ii)socio-economic effect}} + \underbrace{\mathbf{GCE}_{\text{interaction}}}_{\text{iii)interaction effect}} \quad (3)$$

where the interaction term is computed according to Eq. (4):

$$\mathbf{GCE}_{\text{interaction}} = \mathbf{GCE} - \mathbf{GCE}_{\text{fixedGDP}} - \mathbf{GCE}_{\text{fixedClimate}}. \quad (4)$$

For reporting, the socio-economic and interaction contributions are typically combined to summarize the total effect of economic development. Multi-model means summarize central tendencies across climate models.

## Data availability

All original CMIP6 data[14,44,45] used in this study are publicly available https://esgf-node.ipsl.upmc.fr/search/cmip6-ipsl/. The models and ensemble members used within this study are listed in Supplementary Tables 1–4. The Global gridded GDP under the historical and future scenarios[48] is publicly available https://zenodo.org/records/7898409.

## Code availability

All code necessary to reproduce the analysis is made available on and permanently stored at https://doi.org/10.5281/zenodo.17751353.

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

## Acknowledgements

The authors wish to thank H. Rakotoarimanga, R. Meynadier and K. Frieler for useful discussions, as well as U. Beyerle, R. Lorenz, and L. Brunner for the preparation and maintenance of CMIP6 data. We acknowledge the

World Climate Research Programme, which, through its Working Group on Coupled Modelling, coordinated and promoted CMIP6. We thank the climate modeling groups for producing and making available their model output, the Earth System Grid Federation (ESGF) for archiving the data and providing access, and the multiple funding agencies that support CMIP6 and ESGF. B.B. acknowledges funding received from the European Union's Horizon 2020 research and innovation programme under the Marie Skłodowska-Curie grant agreement No 956396 (European weather extremes: drivers, predictability and impacts (EDIPI) ITN).

## Author contributions

B.B., L.G., and S.I.S. conceptualized the study. The methodology was developed jointly by B.B., L.G., and S.I.S. B.B. performed the formal analysis and conducted the investigation. Project administration and funding acquisition were handled by S.I.S., while supervision was provided by L.G. and S.I.S. B.B. carried out the visualization of results and wrote the original draft of the manuscript. B.B., L.G., and S.I.S. contributed to the review and editing of the manuscript.

## Funding

## Competing interests

The authors declare no competing interests.
