## [Transparent Peer Review file · Nature Communications]

Global Economic Exposure to Climate Change Amplified by Spatially Compounding Climate Extremes

Corresponding Author: Dr Bianca Biess

Version 0:

Reviewer comments:

Reviewer #2

(Remarks to the Author)

This manuscript explores the connection between the increase in spatially compounded extreme events of heat, wet, and dry and global economic risks, analyzing how the cross-regional and planetary-scale dependencies of extreme climate exacerbate regional disparities in economic exposure. This perspective and methodology hold meaningful reference and meaning for socioeconomic decision-making.

However, the research is based on static nominal GDP exposure from 2020, and future GDP may differ significantly from that of 2020. The author should provide evidence to clarify the rationality behind this approach. In addition, the authors must address the following issues:

Major Comments:

1. As the authors state, future GDP exposure to climate extremes will also be modulated by changes in GDP levels associated with future societal development. In fact, changes in future GDP exposure may be influenced by three aspects: climate effects, GDP effects, and climate-GDP interaction effects. The author considers the static nominal GDP of 2020 and examines the impact of future compound extreme risks on GDP exposure. However, this premise must be that future GDP exposure is mainly affected by climate effects. Otherwise, the relevance of the results for social decision-making may be significantly diminished, potentially leading to misjudgments and greater economic losses. The author should clarify this crucial issue.

2. P5 Fig. 1: As for Cross-Regional Concurrent GDP Affected, it is hard to understand the criteria for the selection and arrangement of the regions corresponding to the X-axis and Y-axis. Why are there 22 regions on the Y-axis and 21 regions on the X-axis? Does this seem random? What does the value with add. Fraction of GDP represent when there is no corresponding region on the X-axis and Y-axis? It is easy to understand the methodology, but difficult to understand the results of Cross-Regional Concurrent GDP Affected.

3. In the study, the connections between economic exposure and four extreme events were respectively explored, but there was a lack of comparative discussions on the exposure of different extreme events to GDP. For example, among the four extreme events mentioned, which event might have a greater impact on the economic exposure of low-income countries? Because these four events are interrelated, for instance, heatwaves can cause water scarcity. Therefore, results about water scarcity might be part of the consequences caused by heatwaves.

Minor Comments:

1. P3 L76: For 43 trillion USD of GDP, what is the range of its uncertainty? Similarly, GDP related to other extreme events should also be provided.

2. P3 L90: The Arctic amplification can also have an impact on heatwaves.

3. P11 L277: Why is it a season? Maybe it's a year according to your description?

4. P11 L281, L284: For events that occur once every 20 years, it undershoots 5%. This description is ambiguous and may lead readers to mistakenly believe that it represents a range from the 95th to the 100th.

5. P11 L282: Is the root zone the surface soil layer (0-10 cm)? Please provide references.

6. P11 4.2 GDP affected: Why is the reference time for nightlight intensity data 2016 and that for population data 2020? Are the populations and the nightlight intensity of all grids simultaneously used as weights for GDP exposure? Is there any situation where the nightlight intensity or population data of the grids are missing? If so, how to assign weights?

7. P11 L305: According to what weighting? The author should elaborate on the methods used.

8. Appendix A: There are overlaps between the colorbar and the results. Please modify it.

(Remarks on code availability)

Reviewer #3

(Remarks to the Author)

I've reviewed the manuscript titled "Global Economic Exposure to Climate Change Amplified by Spatially Compounding Climate Extremes." Here are my suggestions and recommendations:

1. Grammar and Corrections

The manuscript is generally well-written, but there are a few areas where grammar and phrasing could be improved:

- Abstract: "Here, we demonstrate a crucial link between the projected increase in spatially compounding hot, wet, and dry extremes and the amplification of global economic exposure." Consider rephrasing for clarity: "We demonstrate a crucial link between the projected increase in spatially compounding hot, wet, and dry extremes and the amplification of global economic exposure."
- Introduction: "Global economic losses due to weather-related events have increased eight-fold from the 1970s to the 2010s reaching a cumulative economic loss of 1.48 trillion USD within the period 2010 - 2019." Suggest adding a comma for readability: "Global economic losses due to weather-related events have increased eight-fold from the 1970s to the 2010s, reaching a cumulative economic loss of 1.48 trillion USD within the period 2010 - 2019."
- Results: "The future additional global annual GDP affected by Global Concurrent Extremes (GCE) — defined as extreme events, such as heatwaves, that occurred on average every 20 years in a preindustrial climate — is projected to increase significantly under enhanced global warming compared to present-day conditions (+1.2°C of global warming)." This sentence is quite long and could be broken into two for better readability.

2. Methodology

The methodology is comprehensive and well-detailed. However, it could benefit from a clearer structure:

- Climate Extreme Indicators: The definitions of heatwaves, heavy precipitation, water scarcity, and agroecological droughts are clear. Consider adding a brief explanation of why these specific indicators were chosen.
- GDP Affected: The process of intersecting climate extreme events with the GDP exposure dataset is well-explained. It might be helpful to include a flowchart or diagram to visually represent this process.
- Calculation of Global Warming Levels: This section is detailed but could be simplified. Consider summarizing the key steps and providing references for readers who want more in-depth information.

3. Data Analysis

The data analysis is thorough and uses appropriate statistical methods. Here are some suggestions:

- Figures and Tables: The figures are informative but could be accompanied by more detailed captions explaining the key findings. For example, Figure 1 could include a brief summary of the main trends observed.
- Regional Disparities: The analysis of regional disparities is insightful. Consider adding more context about why certain regions are more affected and how this aligns with existing literature.

4. Clarity of Results

The results are generally clear, but some sections could be enhanced for better understanding:

- Heatwaves: The discussion on heatwaves is detailed. Consider summarizing the key points at the end of the section to reinforce the main findings.
- Heavy Precipitation, Soil Moisture Drought, and Water Scarcity: These sections are well-explained. Adding a summary table comparing the impacts of these different extremes on GDP could be helpful.

5. Conclusion

The conclusion effectively summarizes the study's findings but could be more impactful:

- Key Takeaways: Highlight the most significant findings and their implications for policy and future research.
- Recommendations: Provide specific recommendations for policymakers and researchers based on the study's results.

Decision: Major Revision

Given the thoroughness of the study and the relatively minor issues identified, I would recommend a Major revision.

(Remarks on code availability)

I've reviewed the manuscript titled "Global Economic Exposure to Climate Change Amplified by Spatially Compounding Climate Extremes." Here are my suggestions and recommendations:

1. Grammar and Corrections

The manuscript is generally well-written, but there are a few areas where grammar and phrasing could be improved:

- Abstract: "Here, we demonstrate a crucial link between the projected increase in spatially compounding hot, wet, and dry extremes and the amplification of global economic exposure." Consider rephrasing for clarity: "We demonstrate a crucial link between the projected increase in spatially compounding hot, wet, and dry extremes and the amplification of global economic exposure."
- Introduction: "Global economic losses due to weather-related events have increased eight-fold from the 1970s to the 2010s reaching a cumulative economic loss of 1.48 trillion USD within the period 2010 - 2019." Suggest adding a comma for readability: "Global economic losses due to weather-related events have increased eight-fold from the 1970s to the 2010s, reaching a cumulative economic loss of 1.48 trillion USD within the period 2010 - 2019."
- Results: "The future additional global annual GDP affected by Global Concurrent Extremes (GCE) — defined as extreme events, such as heatwaves, that occurred on average every 20 years in a preindustrial climate — is projected to increase significantly under enhanced global warming compared to present-day conditions (+1.2°C of global warming)." This sentence is quite long and could be broken into two for better readability.

2. Methodology

The methodology is comprehensive and well-detailed. However, it could benefit from a clearer structure:

- Climate Extreme Indicators: The definitions of heatwaves, heavy precipitation, water scarcity, and agroecological droughts are clear. Consider adding a brief explanation of why these specific indicators were chosen.
- GDP Affected: The process of intersecting climate extreme events with the GDP exposure dataset is well-explained. It might be helpful to include a flowchart or diagram to visually represent this process.
- Calculation of Global Warming Levels: This section is detailed but could be simplified. Consider summarizing the key steps and providing references for readers who want more in-depth information.

3. Data Analysis

The data analysis is thorough and uses appropriate statistical methods. Here are some suggestions:

- Figures and Tables: The figures are informative but could be accompanied by more detailed captions explaining the key findings. For example, Figure 1 could include a brief summary of the main trends observed.
- Regional Disparities: The analysis of regional disparities is insightful. Consider adding more context about why certain regions are more affected and how this aligns with existing literature.

4. Clarity of Results

The results are generally clear, but some sections could be enhanced for better understanding:

- Heatwaves: The discussion on heatwaves is detailed. Consider summarizing the key points at the end of the section to reinforce the main findings.
- Heavy Precipitation, Soil Moisture Drought, and Water Scarcity: These sections are well-explained. Adding a summary table comparing the impacts of these different extremes on GDP could be helpful.

5. Conclusion

The conclusion effectively summarizes the study's findings but could be more impactful:

- Key Takeaways: Highlight the most significant findings and their implications for policy and future research.
- Recommendations: Provide specific recommendations for policymakers and researchers based on the study's results.

Decision: Major Revision

Given the thoroughness of the study and the relatively minor issues identified, I would recommend a Major revision.

Version 1:

Reviewer comments:

Reviewer #2

(Remarks to the Author)

I appreciate the author for their efforts in validating the rationality of static nominal GDP exposure. This revision has greatly improved the logical coherence of manuscript, thereby elevating its reference value for future governmental policy considerations. Overall, most of the comments have been addressed. I have only two straightforward suggestions. If the author has completed these revisions, I think this manuscript can be considered for publication in Nature Communications.

1.P3 L86: The first appearance of the abbreviation, provide the full form of SSPs.

2.P14 L282-L290: The first paragraph of the discussion section seems more appropriate to be clarified in the introduction section, because this manuscript is based on these existing backgrounds for an in-depth analysis.

(Remarks on code availability)

Reviewer #3

(Remarks to the Author)

Accept

(Remarks on code availability)

Accept

Reviewer #1 (Remarks to the Author):

This manuscript explores the connection between the increase in spatially compounded extreme events of heat, wet, and dry and global economic risks, analyzing how the cross-regional and planetary-scale dependencies of extreme climate exacerbate regional disparities in economic exposure. This perspective and methodology hold meaningful reference and meaning for socioeconomic decision-making.

However, the research is based on static nominal GDP exposure from 2020, and future GDP may differ significantly from that of 2020. The author should provide evidence to clarify the rationality behind this approach. In addition, the authors must address the following issues:

We thank the reviewer for the thoughtful comments regarding the implications of using static nominal GDP versus time-variant GDP levels. We fully acknowledge the importance of incorporating future GDP trajectories rather than relying solely on static 2020 values. In response, we have substantially revised our analysis and now assess the evolution of GDP exposed to spatially compounding climate extremes across five distinct future GDP scenarios. To maintain a reasonable manuscript length, we present the full scenario trajectories in the global analysis, while the main regional assessment focuses on mid-term (2041–2060) exposure levels under SSP2-4.5, relative to present-day (2001–2020).

Results for additional time horizons and alternative socioeconomic pathways are provided in the Extended Data section

Major Comments:

1. As the authors states, future GDP exposure to climate extremes will also be modulated by changes in GDP levels associated with future societal development. In fact, changes in future GDP exposure may be influenced by three aspects: climate effects, GDP effects, and climate-GDP interaction effects. The author considers the static nominal GDP of 2020 and examines the impact of future compound extreme risks on GDP exposure. However, this premise must be that future GDP exposure is mainly affected by climate effects. Otherwise, the relevance of the results for social decision-making may be significantly diminished, potentially leading to misjudgments and greater economic losses. The author should clarify this crucial issue.

We thank the reviewer for the detailed feedback regarding the different drivers. To address this, we now incorporate projected GDP trajectories and compute the fraction of GDP exposed under time-varying GDP conditions (revised Figure 2b/4a/5a/6a). For comparison, we also perform calculations using static nominal GDP from 2020 to isolate the impact of climate change alone. As expected, the absolute amount of GDP exposed is substantially higher when accounting for projected GDP growth, reflecting GDP increases under all SSP–RCP scenarios (see Figure 1b). However, the relative fraction of GDP exposed is only marginally affected by GDP growth. This is illustrated in the bar plots of revised Figures 2b/4a/5a/6, which explicitly separate the contributions from changes in extreme event occurrence (“climate effects”) and from GDP growth (“socio-economic effects”, encompassing both GDP growth effects and climate–GDP interaction effects). Across all climate extremes investigated, the dominant driver of changes in relative global economic exposure is climate-induced variation in extreme event occurrence, although the magnitude of this effect varies by event type and region. Extended Figure A3 provides the same decomposition at the regional level.

2.P5 Fig.1: As for Cross-Regional Concurrent GDP Affected, it is hard to understand the criteria for the selection and arrangement of the regions corresponding to the X-axis and Y-axis. Why are there 22 regions on the Y-axis and 21 regions on the X-axis? Does this seem random? What does the value with add. Fraction of GDP represent when there is no corresponding region on the X-axis and Y-axis? It is easy to understand the methodology, but difficult to understand the results of Cross-Regional Concurrent GDP Affected.

We thank the reviewer for pointing this out. In the original version, we labeled only every second region on the x-axis (and the corresponding opposite on the y-axis) to maintain a readable font size. We agree that this choice may have caused confusion regarding the selection and arrangement of regions.

To address this, we have revised Figures 3–6 (and Extended Figures A4-A29) in the updated manuscript so that all regions are now displayed on both axes. This improves clarity and allows for a more intuitive interpretation of the Cross-Regional Concurrent GDP Affected results.

3.In the study, the connections between economic exposure and four extreme events were respectively explored, but there was a lack of comparative discussions on the exposure of different extreme events to GDP. For example, among the four extreme events mentioned, which event might have a greater impact on the economic exposure of low-income countries? Because these four events are interrelated, for instance, heatwaves can cause water scarcity. Therefore, results about water scarcity might be part of the consequences caused by heatwaves.

We thank the reviewer for this thoughtful and constructive comment. In response, we have expanded the discussion section (Lines 282–368 in the updated manuscript) to provide a broader reflection on the historical economic impacts of observed extreme events, highlighting which regions have been disproportionately affected and through which impact channels. We also compare these insights with our key findings on projected regional disparities and their amplification through the global and regional co-occurrence of events.

We refrain from drawing definitive conclusions about which extreme event type might have the greatest economic impact in specific regions or income groups, as our analysis is based on exposure assessments rather than impact assessments. Establishing causal economic damages would require explicit modeling of vulnerability and adaptive capacity, which is beyond the scope of this study. Our approach is therefore designed to describe changes in exposure without inferring impacts that cannot be directly derived from our results. To avoid potential misunderstandings, we have clarified more explicitly throughout the manuscript that our analysis focuses on exposure extreme events rather than GDP impacts of extreme events. For example, we revised the title of Section 2.1 in the updated manuscript from “Impacts” to “Exposure” to better reflect this distinction.

We also now emphasize that although the four extreme event types can be correlated (e.g., heat extremes contributing to water scarcity), we assess their effects on economic exposure individually. To prevent misinterpretation, we explicitly state in the updated manuscript (L297-303) that we investigate univariate extremes and that exposure estimates for different event types should not be interpreted as additive. Finally, to account for potential overlaps in economic exposure to different extreme events, we have added Figure A4 to the Extended Figures, which shows the exposure of GDP to the occurrence of at least one extreme event type (i.e., heatwave and/or heavy precipitation and/or water scarcity and/or soil moisture drought).

Minor Comments:

1.P3 L76: For 43 trillion USD of GDP, what is the range of its uncertainty? Similarly, GDP related to other extreme events should also be provided.

We added the 5th–95th percentile uncertainty ranges to all reported values.

2.P3 L90: The Arctic amplification can also have an impact on heatwaves.

We thank the reviewer for pointing this out. Our initial statement was intended to convey that, despite the phenomenon of Arctic amplification, tropical regions exhibit particularly enhanced exposure, even though one might expect the high-latitude regions to show the highest exposure. We did not mean to imply that Arctic amplification has no effect on tropical exposure, but we understand how this could have been misinterpreted. We have therefore decided to remove this sentence for clarity.

3.P11 L277: Why is it a season? Maybe it's a year according to your description?

We thank the reviewer for spotting this detail. We apologize for the confusion, as we intended to refer to a year in which a heavy precipitation event occurs, not a season associated with a wet spell. We have revised the manuscript accordingly.

4.P11 L281, L284: For events that occur once every 20 years, it undershoots 5%. This description is ambiguous and may lead readers to mistakenly believe that it represents a range from the 95th to the 100th.

We thank the reviewer for pointing this out. We agree that the original phrasing was ambiguous. To clarify, we calculated the undershoot of the 5th percentile of annual soil moisture drought (MRSOS) or $P-E$ (for water scarcity) monthly minima. We have revised the text accordingly to avoid any potential misinterpretation.

5.P11 L282: Is the root zone the surface soil layer (0-10 cm)? Please provide references.

Yes, the root zone of the surface layer refers to the top 10 cm of the soil. We have added this clarification to the manuscript.

6.P11 4.2GDP affected: Why is the reference time for nightlight intensity data 2016 and that for population data 2020? Are the populations and the nightlight intensity of all grids simultaneously used as weights for GDP exposure? Is there any situation where the nightlight intensity or population data of the grids are missing? If so, how to assign weights?

We thank the reviewer for raising this point. As we no longer rely on the global high-resolution GDP exposure dataset provided by Eberenz et al. (2020), this comment is no longer relevant. In the revised manuscript, we now refer to the global gridded GDP dataset consistent with the Shared Socioeconomic Pathways (SSPs).

7.P11 L305: According to what weighting? The author should elaborate on the methods used.

We thank the reviewer for pointing out the missing description. Due to our revised calculation approach, we no longer rely on global mean temperatures. Therefore, we consider this comment no longer relevant.

8. Appendix A: There are overlaps between the colorbar and the results. Please modify it.
We thank the reviewer for pointing out this issue. Based on our updated calculations, we replotted the data and ensured that no overlaps remain.

Reviewer #2 (Remarks to the Author):

I've reviewed the manuscript titled "Global Economic Exposure to Climate Change Amplified by Spatially Compounding Climate Extremes." Here are my suggestions and recommendations:

1. Grammar and Corrections

The manuscript is generally well-written, but there are a few areas where grammar and phrasing could be improved:

- **Abstract:** "Here, we demonstrate a crucial link between the projected increase in spatially compounding hot, wet, and dry extremes and the amplification of global economic exposure." Consider rephrasing for clarity: "We demonstrate a crucial link between the projected increase in spatially compounding hot, wet, and dry extremes and the amplification of global economic exposure."
We thank the reviewer for the suggestion. We have updated the sentence accordingly.
- **Introduction:** "Global economic losses due to weather-related events have increased eight-fold from the 1970s to the 2010s reaching a cumulative economic loss of 1.48 trillion USD within the period 2010 - 2019." Suggest adding a comma for readability: "Global economic losses due to weather-related events have increased eight-fold from the 1970s to the 2010s, reaching a cumulative economic loss of 1.48 trillion USD within the period 2010 - 2019."
We have made the corresponding change.
- **Results:** "The future additional global annual GDP affected by Global Concurrent Extremes (GCE) — defined as extreme events, such as heatwaves, that occurred on average every 20 years in a preindustrial climate — is projected to increase significantly under enhanced global warming compared to present-day conditions (+1.2°C of global warming)." This sentence is quite long and could be broken into two for better readability.
We revised the sentence and broke it into two parts to enhance clarity.

2. Methodology

The methodology is comprehensive and well-detailed. However, it could benefit from a clearer structure:

- **Climate Extreme Indicators:** The definitions of heatwaves, heavy precipitation, water scarcity, and agroecological droughts are clear. Consider adding a brief explanation of why these specific indicators were chosen.
We thank the reviewer for this suggestion. The selected indicators (heatwaves, heavy precipitation, water scarcity, and agroecological drought) follow definitions used in the IPCC

Sixth Assessment Report and were chosen because they represent well-understood manifestations of anthropogenic climate change with high societal relevance across key sectors, including health, agriculture, and infrastructure. We added this explanation to the manuscript (L445-L449).

- **GDP Affected: The process of intersecting climate extreme events with the GDP exposure dataset is well-explained. It might be helpful to include a flowchart or diagram to visually represent this process.**

We thank the reviewer for this helpful suggestion. A flowchart has been added to the main text (Figure 1 in the revised manuscript) to visually illustrate how climate extreme events are intersected with the GDP exposure dataset, thereby clarifying the methodological workflow.

- **Calculation of Global Warming Levels: This section is detailed but could be simplified. Consider summarizing the key steps and providing references for readers who want more in-depth information.**

We thank the reviewer for the suggestion. We have revised our analysis to compare the near-term (2041–2060) and far-term (2081–2100) periods with the present-day period (2001–2020), instead of describing results in terms of global warming levels. Consequently, the section on calculating global warming levels has been removed.

3. Data Analysis

The data analysis is thorough and uses appropriate statistical methods. Here are some suggestions:

- **Figures and Tables: The figures are informative but could be accompanied by more detailed captions explaining the key findings. For example, Figure 1 could include a brief summary of the main trends observed.**

We thank the reviewer for this valuable suggestion. Due to journal-imposed caption length limits, extended summaries cannot be added directly to figure captions. Instead, we have added a concise summary paragraph at the end of each variable section to reinforce the key trends and improve readability, as also recommended by the reviewer.

- **Regional Disparities: The analysis of regional disparities is insightful. Consider adding more context about why certain regions are more affected and how this aligns with existing literature.**

We thank the reviewer for this helpful suggestion. We expanded the discussion of regional disparities in two ways:

- *Results section: We investigate which driver (socio-economic effects or climate effects) primarily drives the additional GDP exposure and compare regional differences.*
- *Discussion section: For each extreme event type, we now provide a summary of existing literature on which regions are expected to be most affected due to human-induced climate change. We then compare this knowledge to our results by showing where projected increases in extreme event occurrence are reflected in our analysis of heightened GDP exposure. This explicitly connects our findings to the broader literature and contextualizes regional differences.*

4. Clarity of Results

The results are generally clear, but some sections could be enhanced for better understanding:

- **Heatwaves:** The discussion on heatwaves is detailed. Consider summarizing the key points at the end of the section to reinforce the main findings.

We added a summary paragraph at the end of each extreme event section (including heatwaves) to highlight the key takeaways. We agree that this improves clarity and helps the reader to focus on the main findings.

- **Heavy Precipitation, Soil Moisture Drought, and Water Scarcity:** These sections are well-explained. Adding a summary table comparing the impacts of these different extremes on GDP could be helpful.

We thank the reviewer for this suggestion. As noted above, we have added summary paragraphs for all extreme event types. To avoid overloading the main text, we have not included a summary table, as our study focuses primarily on the qualitative implications of spatially compounding climate extremes. However, should the reviewer consider the table essential, we would be pleased to include it in a revised version or as supplementary material.

5. Conclusion

The conclusion effectively summarizes the study's findings but could be more impactful:

- **Key Takeaways:** Highlight the most significant findings and their implications for policy and future research.
- **Recommendations:** Provide specific recommendations for policymakers and researchers based on the study's results.

We thank the reviewer for this valuable suggestion. As noted above, we have summarized the key findings for each extreme event type within their respective sections. In addition, we have expanded the discussion and included a dedicated closing paragraph that synthesizes the main insights of the study (see Lines 404–418 in the revised manuscript). While our work aims to provide policy-relevant evidence, we intentionally refrain from making prescriptive recommendations for policymakers. The primary objective of this study is to deliver robust scientific evidence that can inform decision-making without imposing specific policy directives.

Decision: Major Revision

Given the thoroughness of the study and the relatively minor issues identified, I would recommend a Major revision.

We appreciate the reviewer's careful assessment and constructive feedback, which have substantially improved the clarity and quality of our manuscript.

Response to Referee #2

I appreciate the author for their efforts in validating the rationality of static nominal GDP exposure. This revision has greatly improved the logical coherence of manuscript, thereby elevating its reference value for future governmental policy considerations. Overall, most of the comments have been addressed. I have only two straightforward suggestions. If the author has completed these revisions, I think this manuscript can be considered for publication in Nature Communications.

We thank the reviewer for thoroughly reading the manuscript again and for providing valuable input throughout the review process.

1.P3 L86: The first appearance of the abbreviation, provide the full form of SSPs.

The first appearance of SSPs has been clarified with the full form (Shared Socioeconomic Pathways). Other abbreviations have been updated accordingly.

2.P14 L282-L290: The first paragraph of the discussion section seems more appropriate to be clarified in the introduction section, because this manuscript is based on these existing backgrounds for an in-depth analysis.

We thank the reviewer for this suggestion. Accordingly, we have moved the first paragraph of the Discussion section to the Introduction and clarified it to better provide the background for the in-depth analysis presented in this study.

Response to Referee #3

Accept

We thank the reviewer for the positive feedback and for providing valuable input throughout the review process.